# Dynamic Analysis and Piezoelectric Energy Harvesting from a Nonideal Portal Frame System including Nonlinear Energy Sink Effect

**Angelo M. Tusset** [1,*] , **Dim B. Pires** [1], **Jose M. Balthazar** [1] , **Maria E. K. Fuziki** [2], **Dana I. Andrade** [1] **and Giane G. Lenzi** [1]

1    Department of Production Engineering, Federal University of Technology-Paraná, Paraná-Doutor Washington Subtil Chueire St. 330, Ponta Grossa 84017-220, Brazil; dim.piress@gmail.com (D.B.P.); jmbaltha@gmail.com (J.M.B.); dana@alunos.utfpr.edu.br (D.I.A.); gianeg@utfpr.edu.br (G.G.L.)
2    Department of Chemical Engineering, State University of Maringá, Colombo Ave. 5790, Maringá 87020-900, Brazil; mariafuziki@alunos.utfpr.edu.br
*    Correspondence: tusset@utfpr.edu.br; Tel.: +55-42-3220-4800

**Abstract:** This paper investigates, through numerical simulations, the application of piezoelectric materials in energy generation. The mathematical model describes a U-shaped portal frame system, excited by an engine with unbalanced mass and coupled to a nonlinear energy sink (NES), which is used as a passive vibration absorber. The influence of the piezoelectric material parameters used in the energy collection and the dimensioning parameters of the NES system is deeply analyzed in this paper. Numerical simulations are presented considering all combinations of the parameters of the piezoelectric material model and the NES. The system dynamics were analyzed through phase diagrams and the 0–1 test. The estimation of energy collection was carried out by calculating the average power. The numerical results show that a more significant potential for energy generation is obtained for certain combinations of parameters, as well as chaotic behavior in some cases.

**Keywords:** piezoelectric material; passive absorbers; electromechanical systems





## 1. Introduction

Electric energy is essential for technological development, economic growth, and population life quality improvement. Fossil fuels, such as oil, coal, and natural gas, are still the most used sources for electricity production. However, the use of fossil fuels is related to the increase in the greenhouse gases concentration in the atmosphere, which is responsible for environmental problems, such as air pollution and climate change [1,2].

As nonrenewable resources, fossil fuels are finite. However, estimates indicate that global demand in the mid-21st century will exceed the available reserves of fossil fuels [3]. The concern with energy security and environmental protection highlighted the need to produce electricity from renewable sources, such as water, the sun, wind, biomass, and geothermal energy [4,5].

Energy harvesting is a promising technology that sustainably contributes to electricity production, capturing wasted or unused energy from the environment. Energy can be captured from various sources, such as solar irradiation, wind, vibration, heat, and water flow. The main advantage of energy harvesting is that it is not dependent on weather and environmental conditions like solar, wind, and thermal energy. In addition, energy harvesters can be installed in places often subjected to mechanical stress, such as bridges or highways. In this way, collecting clean and renewable energy is a sustainable alternative that alleviates the environmental impacts generated by energy production from fossil fuels [6–9].

In the current era, Internet of Things (IoT) devices are essential for controlling and collecting information. They are applied in the most diverse areas, such as biomedicine

and health, the environment, public safety, and industry. Thus, the demand for electronic devices has been rising recently, so there are concerns about the power supply to these devices. The life of the batteries is limited, and recharging or replacing them can be difficult in some cases due to the human effort and cost involved. In this context, energy harvesters are presented as an efficient and ecological solution to this challenge, as they capture and store energy for electronic devices [10–12]. There are several energy harvesting systems, such as piezoelectric, triboelectric, thermoelectric, pyroelectric, and photovoltaic systems and energy harvesting systems based on water evaporation [13].

Among the energy harvesting methods, the present work opted for the vibrational collection method using piezoelectric materials. According to [14], research on vibration energy collectors arouses great interest and is widely cited in the literature. Ref. [15] stated that harvesting vibrational energy via piezoelectric materials has the advantages of a good configuration, electromechanical conversion efficiency, and output power.

Piezoelectric materials can convert mechanical vibrations into electrical voltage, a property also called piezoelectricity. The process of the application of mechanical stress to the material producing electrical energy is called the direct piezoelectric effect. In contrast, the process in which mechanical deformation occurs in the material because of an applied electrical voltage is known as the indirect piezoelectric effect [16,17]. These characteristics make piezoelectric materials extremely versatile, as they can be used as energy collectors, sensors, or actuators. As sensors, they can be used to determine structure deflections. As actuators coupled to a structure, they can transfer the mechanical deformation they suffer when receiving an electrical voltage to the structure.

Since their discovery, piezoelectric materials have been applied in advanced technologies in diverse areas, like energy harvesting, given their ability to sense and collect vibrations [18,19].

The study developed by [20] proposed a piezoelectric energy harvester for rotating machines, which can not only harvest mechanical energy from the bending deformation of a rotating shaft but also has the capability of detecting typical rotor faults, such as crack and rub-impact faults. Hence, the proposed system can be used to power wireless sensors, as well as a self-powered sensor to monitor the rotating machine's conditions. A piezoelectric energy harvester to collect the energy generated by pedestrian movement was proposed by [21]. The harvested energy can be employed to power LED lights and charge mobile devices. The study conducted by [22] developed a piezoelectric energy harvester for smart pavements. The proposed system can power low-power electronics, self-powered sensors, and remote electrical equipment in transportation infrastructures. A vibro-impact piezoelectric energy harvester for low-frequency vibration enhanced by acoustic black holes (ABHs) was presented in the work conducted by [23]. The proposed energy harvester has two beams that collide with each other, and the impact of this collision can transfer the vibration energy from the low-frequency band to the high-frequency band, where the ABH obtains a desirable energy focalization effect to improve the output performance of the energy harvester. Lead-free eco-friendly cement-based piezoelectric composites for smart concrete structure applications were designed and manufactured by [24] to harvest the building environment energy, such as vibration energy, impact energy, and wind energy, among others. An energy harvest based on a bistable Origami Mechanism was proposed by [25]. The system presents a compact size, light weight, large deformability, stretchability, and flexibility, which makes the system advantageous for integration into various hosts subjected to vibrations. All-in-one self-charging supercapacitor power cells (SCSPCs) were manufactured by [26]. In the study, graphene-based SCSPC devices using a porous PVDF piezopolymer separator (incorporated with an ionic liquid electrolyte) were proposed for the direct conversion and storage of mechanical energy into electrical energy. The findings of the study contribute to research on the design and development of a single device capable of collecting, converting, and storing energy.

According to [27], the most used type of piezoelectric material in energy collection is piezoelectric ceramics. These materials, however, have a high fragility, limiting the

amount of voltage they withstand without suffering damage. Studies aimed at developing more flexible piezoelectric materials, so-called polymeric piezoelectric materials, have been designed to overcome the limitations of piezoelectric ceramics. Some examples of polymeric piezoelectric materials are reported in the literature, such as the flexible composites NBT-BT/PVDF (ceramic filler powders ($[(NaBi)TiO_3] - BaTiO_3$) mixed with PVDF powder) developed by [28], an elastic composite material from polymer powders (polyurethane and polypropylene) with the addition of $BaTiO_3$ proposed by [29], and a piezoelectric energy harvester based on polyvinylidene fluoride with the addition of lithium chloride (PVDF/LiCl) electrospun nanofibers presented by [30]. The literature also presents studies focused on piezoelectric nanogenerators [31,32], which, due to their small size, have applications in emerging technologies of electronics and biology, such as wearable electronic devices, implantable biomedical devices, sensors, and portable electronics [33,34].

Energy harvesting through piezoelectric materials can be a potential solution to the challenges of energy production since it makes it possible to transform environmental energy sources into valuable energy, which until then would be wasted [35]. Some examples are presented in the literature, such as the energy collection from oceans [36], wind [37], rain [38], and road vibration [39]. Energy harvesting by piezoelectric materials has applications in areas such as biomedicine and health [40,41], wireless data transmission [42–44], aeronautics [45–47], environmental monitoring, and artificial intelligence [48], among others.

Vibration mitigation and energy dissipation in mechanical systems is a rapidly developing field, as it is necessary to protect these systems from vibration-related problems. Vibration absorbers protect structural dynamic systems via the passive transfer and dissipation of energy. This occurs through a process called Target Energy Transfer (TET), in which energy is transferred from a primary dynamic structure to an attached receiving dynamic system irreversibly. TET is triggered by the nonlinear interaction caused by the coupling of the nonlinear energy sink (NES) with the structure [49–51].

The nonlinear energy sink (NES) concept was first proposed in 2001 by [52,53]. The NES device is composed of an additional mass component attached to the primary structure, with a highly nonlinear stiffness. In vibration control, the NES presents the advantages of a high robustness, structural simplicity, high reliability, and effective vibration suppression. Therefore, it is increasingly used in different structures and is being applied in research in numerous fields [54–56], including civil engineering [57], mechanical engineering [58–60], aeronautics [61], and energy harvesting [62–65].

According to the results obtained in the studies of [66–69], the use of an absorber combined with a nonlinear energy sink and a piezoelectric energy harvest can simultaneously improve vibration suppression and energy harvesting.

In this context, this paper investigates the application of materials with piezoelectric characteristics for micropower generation. A mathematical model was used to describe a U-portal frame system coupled to a motor in continuous operation and a nonlinear energy sink (NES), acting as a passive vibration absorber. As its main contribution, this paper presents a deep analysis of the influence of the parameters of the piezoelectric material and the NES system coupled to the structure. Through numerical simulations, all combinations of the parameters are presented, and the dynamics of the system and the estimated energy collection for each parametric variation are analyzed. The 0–1 test was applied to examine whether the system behavior is periodic or chaotic and the calculation of the average power for the energy collection estimate.

## 2. Materials and Methods

The analyzed energy collection system is represented by Figure 1, consisting of a U-shaped portal frame structure base with nonlinear rigidity and a nonlinear energy sink (NES), with the piezoelectric material attached to the side of the structure.

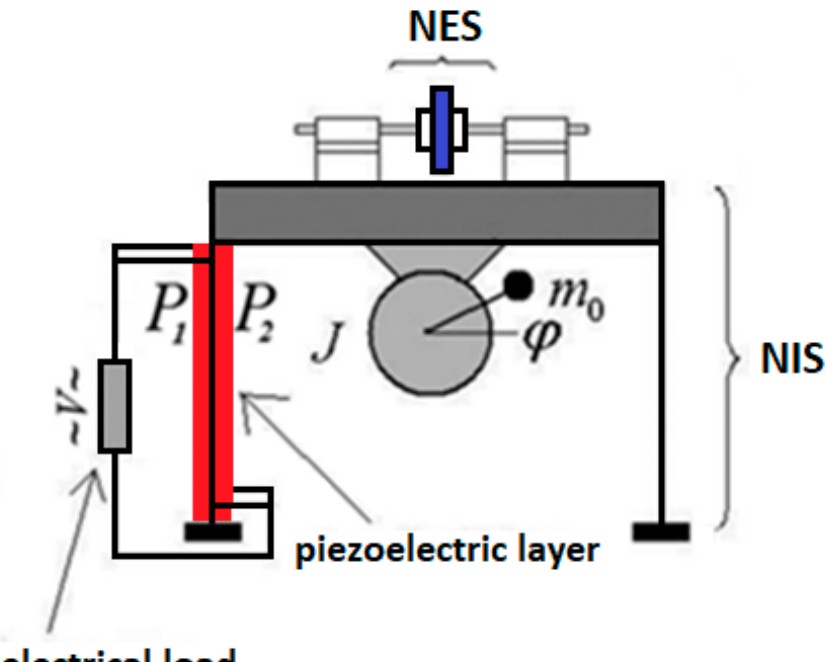

**Figure 1.** U-shaped portal frame structure with NES and coupled piezoelectric material.

The lateral displacements of the structure come from the excitation generated by a direct current (DC) electric motor, which has an unbalanced mass ($m_0$), thus characterizing a nonideal vibrating system (NIS) [70] attached to the structure of an NES. The magnetic components of the NES, as shown in Figure 1, provide, according to [71], cubic rigidity and linear damping to the lateral movements of the mass $m_2$ that slide with its axis in two bearings. And two plates of piezoelectric material are coupled to the structure according to [72].

Figure 2 presents the physical model of the system represented in Figure 1, where $m_0$ is the unbalanced mass of the motor, $m_1$ is the mass of the portal frame, $k$ is the stiffness of the portal frame, $b$ is the damping of the portal frame, $X_1$ is the displacement of the portal frame, $\varphi$ is the angular displacement of the motor, $J$ is the inertia moment of the motor, $r$ is the eccentricity of the unbalanced mass of the motor, $k_1$ is the nonlinear stiffness of the NES, $b_1$ is the damping of the NES, and $X_2$ is the displacement of the NES. The parameter $k$ is composed of two components: $k_l + k_{nl}$, with $k_l$ being the linear stiffness component and $k_{nl}$ the nonlinear stiffness component.

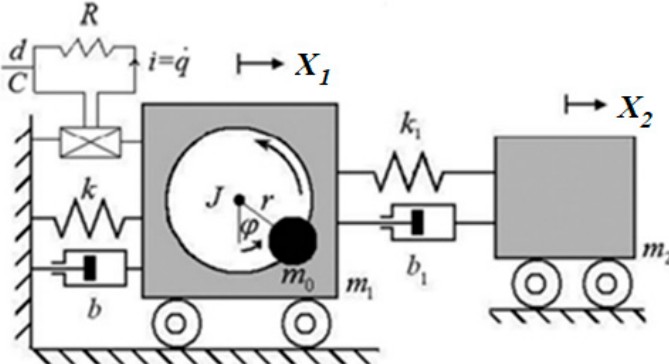

**Figure 2.** Physical model of the U-frame structure with NES and coupled piezoelectric material.

The torque of the motor is a function of the angular and is given by: $V_1 - V_2\dot{\varphi}$, where $V_1$ is related to the voltage applied across the armature of the DC motor and $V_2$ is the constant for each model of DC motor considered [70].

The equation of motion of the electromechanical system represented by Figure 2 is given by the following equation [72–74]:

$$
\begin{aligned}
&(m_1 + m_0)\ddot{X}_1 + b\dot{X}_1 + b_1\left(\dot{X}_1 - \dot{X}_2\right) - k_l X_1 + k_{nl} X_1^3 + k_1\left(X_1 - X_2\right)^3 = \\
&m_0 r\left(\ddot{\varphi}\sin(\varphi) + \dot{\varphi}^2\cos(\varphi)\right) + \frac{d(X_1)}{C}q \\
&m_2\ddot{X}_2 - b_1\left(\dot{X}_1 - \dot{X}_2\right) - k_1\left(X_1 - X_2\right)^3 = 0 \\
&\left(J + r^2 m_0\right)\ddot{\varphi} - rm_0\ddot{X}_1\sin(\varphi) = V_1 - V_2\dot{\varphi} \\
&R\dot{q} - \frac{d(X_1)}{C}X_1 + \frac{q}{C} = 0
\end{aligned} \tag{1}
$$

According to [75], the electrical charge developed in the coupled circuit is given by $q$, and the term $\frac{d(X_1)}{C}q$ represents the piezoelectric coupling to the mechanical component, with a strain-dependent coupling coefficient $d(x)$.

The system represented by Equation (1) can be rewritten in a dimensionless form considering the following substitutions [72–74]: $\tau = \omega_1 t$, $x = \frac{rX_1}{\omega_1^2}$, $z = \frac{rX_2}{\omega_1^2}$,

$\omega_1 = \sqrt{\left(\frac{k_l}{(m_1+m_0)}\right)}$, $\alpha_1 = \frac{b}{(m_1+m_0)\omega_1}$, $\beta_1 = \frac{k_l}{(m_1+m_0)\omega_1^2}$, $\beta_3 = \frac{k_{nl}r^2}{(m_1+m_0)\omega_1^6}$, $\delta_1 = \frac{m_0\omega_1^2}{(m_1+m_0)}$,

$\rho_1 = \frac{m_0 r^2}{(J+r^2 m_0)\omega_1^2}$, $\rho_2 = \frac{V_1}{(J+r^2 m_0)\omega_1^2}$, $\rho_3 = \frac{V_2}{(J+r^2 m_0)\omega_1}$, $\alpha_2 = \frac{b_1}{m_2\omega_1}$, $\alpha_3 = \frac{k_1 r^2}{m_1}$, $\varepsilon_1 = \frac{m_1 b_1}{m_2\omega_1}$,

$\varepsilon_2 = k_1 r^2$, $v = \frac{q}{q_0}$, $]\rho = \frac{RC}{\omega_1}$, and $d(X_1) = \theta(1 + \Theta|x|)$.

Considering the substitutions above, system 1 can be represented in the following dimensionless form:

$$
\begin{aligned}
&\ddot{x} - \beta_1 x + \alpha\dot{x} + \alpha_2\left(\dot{x} - \dot{z}\right) + \beta_3 x^3 + \alpha_3\left(x - z\right)^3 - \theta(1 + \Theta|x|)v = \delta_1\ddot{\varphi}\sin\varphi + \delta_1\dot{\varphi}^2\cos]\varphi \\
&\ddot{z} - \varepsilon_1\left(\dot{x} - \dot{z}\right) - \varepsilon_2\left(x - z\right)^3 = 0 \\
&\ddot{\varphi} = \rho_1\cos\varphi\ddot{x} - \rho_3\dot{\varphi} + \rho_2 \\
&\rho\dot{v} - \theta(1 + \Theta|x|)x + v = 0
\end{aligned} \tag{2}
$$

Using new variables defined as $x_1 = x$, $x_2 = \dot{x}$, $x_3 = z$, $x_4 = \dot{z}$, $x_5 = \varphi$, $x_6 = \dot{\varphi}$, and $x_7 = v$, the equations may be rewritten in a state space representation as follows:

$$
\begin{aligned}
&\dot{x}_1 = x_2 \\
&\dot{x}_2 = \frac{1}{\Delta}\left(\begin{array}{l}\beta_1 x_1 - \alpha_1 x_2 - \alpha_2(x_2 - x_4) - \beta_3 x_1^3 - \alpha_3(x_1 - x_3)^3 + \\ +\theta(1 + \Theta|x_1|)x_7 + \delta_1\cos(x_5)x_6^2 + \delta_1\sin(x_5)(-\rho_3 x_6 + \rho_2)\end{array}\right) \\
&\dot{x}_3 = x_4 \\
&\dot{x}_4 = \varepsilon_1(x_2 - x_4) + \varepsilon_2(x_1 - x_3)^3 \\
&\dot{x}_5 = x_6 \\
&\dot{x}_6 = \frac{1}{\Delta}\left(\begin{array}{l}-\rho_3 x_6 + \rho_2 + \rho_1\cos(x_5)\left(\beta_1 x_1 - \alpha_1 x_2 - \alpha_2(x_2 - x_4) - \beta_3 x_1^3 - \right. \\ -\alpha_3(x_1 - x_3)^3 + \theta(1 + \Theta|x_1|)x_7 + \delta_1\cos(x_5)x_6^2)\end{array}\right) \\
&\dot{x}_7 = (\theta(1 + \Theta|x_1|)x_1 - x_7)/\rho
\end{aligned} \tag{3}
$$

where $\Delta = 1 - \rho_1\cos(x_5)\delta_1\sin(x_5)$.

The average power is obtained through the formula below [74]:

$$
P_{avg} = \frac{1}{T}\int_0^T P(\tau)d\tau \tag{4}
$$

where instantaneous power is calculated using the following:

$$P = \rho \dot{v}^2 \tag{5}$$

For analysis of the periodic or chaotic behavior of the system, the 0–1 test proposed by [76,77] is considered, consisting of estimating a parameter $K$. The test assesses a system variable $x_j$, where two new coordinates $(p, q)$ are defined as follows [78,79]:

$$p\left(n, \bar{c}\right) = \sum_{j=0}^{n} x_j cos(j\bar{c}) \tag{6}$$

$$q\left(n, \bar{c}\right) = \sum_{j=0}^{n} x_j sin(j\bar{c}) \tag{7}$$

where $\bar{c} \in (0, \pi)$ is a constant. The mean square displacement of the new variables $p\left(n, \bar{c}\right)$ and $q\left(n, \bar{c}\right)$ are given by the following [78,79]:

$$M(n, c) = \lim_{n \to \infty} \frac{1}{N} \sum_{j=1}^{N} \left[ \left( p\left(j+n, \bar{c}\right) - p\left(j, \bar{c}\right) \right)^2 + \left( q\left(j+n, \bar{c}\right) - q\left(j, \bar{c}\right) \right)^2 \right] \tag{8}$$

where $n = 1, 2, \ldots, N$ and, therefore, it is obtained the parameter $K_c$ in the limit of a very long time [78,79]:

$$K_c = \frac{cov\left(Y, M(\bar{c})\right)}{\sqrt{var(Y)var(M(\bar{c}))}} \tag{9}$$

where $Y = [1, 2, \ldots, n_{max}]$.

Given any two vectors $x$ and $y$, the covariance $cov(x, y)$ and variance $var(x)$ of $n_{max}$ elements are usually defined as follows [78,79]:

$$cov(x, y) = \frac{1}{n_{max}} \sum_{n=1}^{n_{max}} ((x(n) - \bar{x})((y(n) - \bar{y}))) \tag{10}$$

$$var(x) = cov(x, y) \tag{11}$$

where $\bar{x}$ is the average of $x(n)$ and $\bar{y}$ is the average of $y(n)$.

The value of the $K$ is obtained by taking the median of 100 different values of the parameter $K_c$, considering that the system is periodic for $K$ values close to 0 and the system is chaotic if the $K$ value is close to 1 [78,79].

For numerical simulations, Equation (3) was considered, integrated by the 4th-order Runge–Kutta method, with integration step ($h = 0.01$), variation in parameters $\varepsilon_1$, $\varepsilon_2$, $\theta$, $\Theta$, and $\rho$ and the following fixed parameters: $\alpha_1 = 0.1$, $\alpha_2 = 0.1$, $\alpha_3 = 0.5$, $\beta_1 = 1$, $\beta_3 = 0.2$, $\delta_1 = 8.373$, $\rho_1 = 0.05$, $\rho_2 = 100$, $\rho_3 = 200$, $\varepsilon_1 = 1$, $\varepsilon_2 = 5$, $\theta = 0.20$, $\Theta = 0.60$, and $\rho = 1$ [72,74].

The calculation of the average power will be obtained through Equation (4). The behavior analysis is performed by analyzing the variations in $K$ from test 0–1, according to Equation (9). This paper will consider chaotic behavior for the system when $K \geq 0.8$, undefined behavior when $0.3 \leq K \leq 0.79$, and periodic behavior when $K < 0.3$.

## 3. Results

In this section, analyses of the system dynamics and their potential for energy generation will be presented, considering the following control variables: $\varepsilon_1 = [0.01 : 2]$, $\varepsilon_2 = [1 : 10]$, $\theta = [0.01 : 0.4]$, $\Theta = [0.01 : 1.2]$, and $\rho = [0.01 : 2]$.

### 3.1. Dynamics and Potential of Energy Generation for Variation in Parameters $\varepsilon_1$ and $\varepsilon_2$

Figure 3 presents the $K$ variations in the 0–1 test and the variations in the average power when setting the parameters $\theta = 0.20$, $\Theta = 0.6$, and $\rho = 1.0$ and considering $\varepsilon_1 = [0.01 : 2]$ versus $\varepsilon_2 = [1 : 10]$.

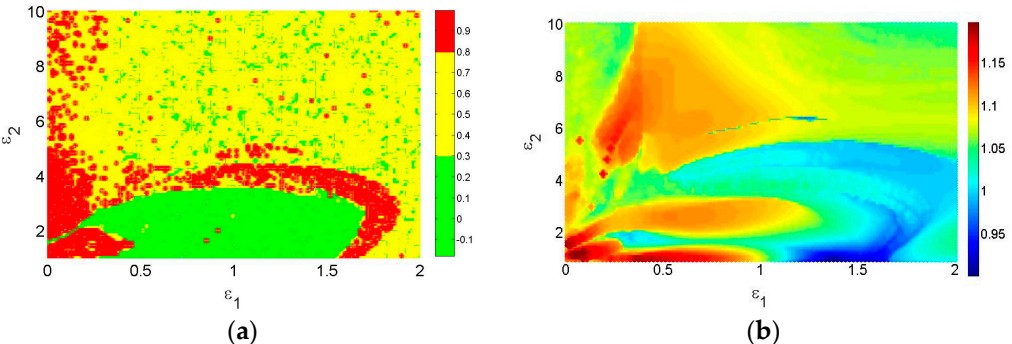

(a)        (b)

**Figure 3.** Variation in parameters $\varepsilon_1 = [0.01 : 2]$ versus $\varepsilon_2 = [1 : 10]$: (**a**) test 0–1; (**b**) average power.

According to the results presented in Figure 3b, the greatest potential for energy generation is obtained when parameters close to $\varepsilon_1 = 0.0502$ and $\varepsilon_2 = 1.182$ are considered, providing an average power estimate $P_{avg} \approx 1.198$. It can also be observed in Figure 3a that the lateral displacements for these parameters become chaotic with the value of $K \approx 0.9$.

However, analyzing the results of Figure 3, it is observed that for $\varepsilon_1 = 0.3919$ and $\varepsilon_2 = 1$ there is also a potential for energy generation in regions of maximums, with an estimated average power of $P_{avg} \approx 1.184$. However, in this case the lateral displacements have periodic behavior, with $K \approx 0$.

### 3.2. Dynamics and Potential of Energy Generation for Variation in Parameters $\varepsilon_1$ and $\theta$

In Figure 4, you can see the $K$ variations in the 0–1 test and the variations in the average power when fixing the parameters $\varepsilon_2 = 5$, $\Theta = 0.6$, and $\rho = 1.0$ and considering $\varepsilon_1 = [0.01 : 2]$ versus $\theta = [0.01 : 0.4]$.

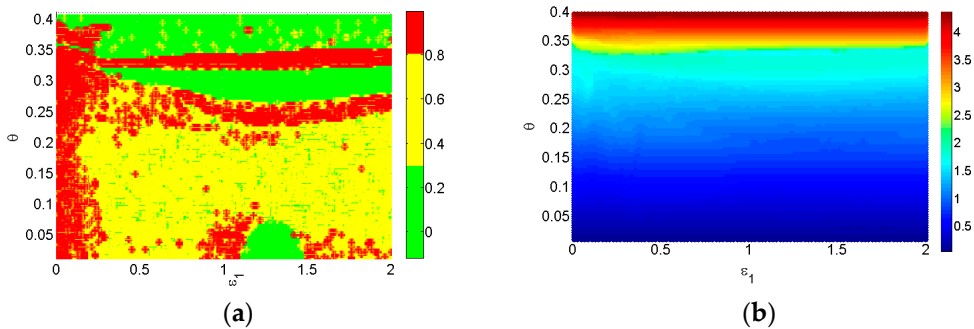

(a)        (b)

**Figure 4.** Variation in parameters $\varepsilon_1 = [0.01 : 2]$ versus $\theta = [0.01 : 0.4]$: (**a**) test 0–1; (**b**) average power.

As shown in Figure 4, the potential energy increases as the value of $\theta$ increases. We can also observe that the highest energy potentials are in the region where the lateral displacement is periodic. So, a higher energy generation value is obtained for the parameters $\varepsilon_1 = 0.0502$ and $\theta = 0.3961$, with $P_{avg} \approx 4.238$ and $K \approx 0$.

### 3.3. Dynamics and Potential of Energy Generation for Variation in Parameters $\varepsilon_1$ and $\Theta$

Setting the parameters $\varepsilon_2 = 5$, $\theta = 0.20$, and $\rho = 1.0$ and performing variations in the parameters $\varepsilon_1 = [0.01 : 2]$ versus $\Theta = [0.01 : 1.2]$, the results shown in Figure 5 are obtained.

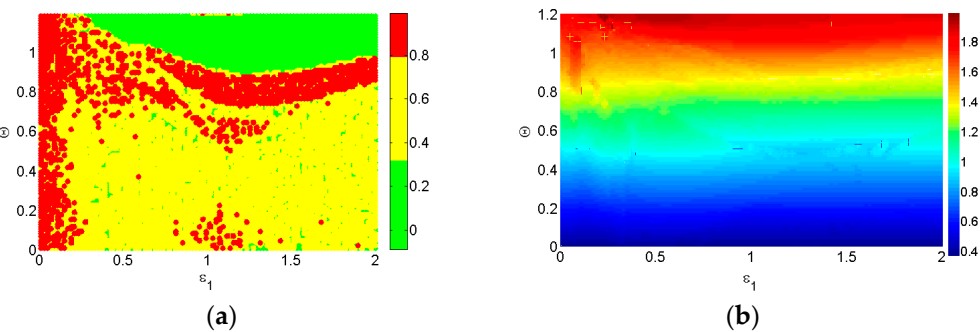

**Figure 5.** Variation in parameters $\varepsilon_1 = [0.01 : 2]$ versus $\Theta = [0.01 : 1.2]$: (**a**) test 0–1. (**b**) average power.

According to the analysis of the results presented in Figure 5, the value of the average power increases as the value of $\Theta$ increases, and the region of the most significant potential for energy generation is also in the region of periodic behavior of the system. This behavior is similar to that observed for the parameter $\theta$. So, we obtain an average power $P_{avg} \approx 1.975$, with $K \approx 0$, for the parameters $\varepsilon_1 = 0.2311$ and $\Theta = 1.188$. It is also possible to observe that a small strip of the chaotic behavior region presents a high energy value, since we have an average power of $P_{avg} \approx 1.587$, with $K \approx 0.9$, for the parameters $\varepsilon_1 = 0.01$ and $\Theta = 1.152$.

### 3.4. Dynamics and Potential of Energy Generation for Variation in Parameters $\varepsilon_1$ and $\rho$

In Figure 6, the variation in the average power of energy generation and the $K$ parameter of test 0–1 are presented, considering variations in $\varepsilon_1 = [0.01 : 2]$ versus $\rho = [0.01 : 2]$, with fixed values for $\alpha_2 = 0.1$, $\theta = 0.20$, and $\Theta = 0.6$.

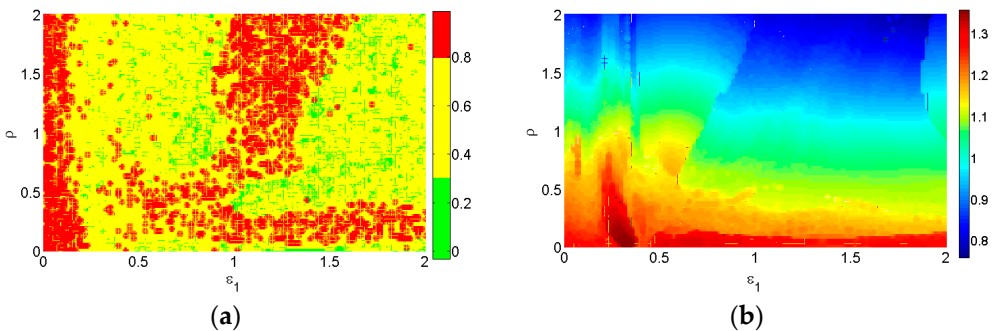

**Figure 6.** Variation in parameters $\varepsilon_1 = [0.01 : 2]$ versus $\rho = [0.01 : 2]$; (**a**) test 0–1; (**b**) average power.

Figure 6 shows that we have a more significant potential for energy generation for lower values of $\rho$, taking the system to defined regions with quasi-periodic behavior. Thus, in the case of $\varepsilon_1 = 0.03316$ and $\rho = 0.01$, an average power $P_{avg} \approx 1.355$ is obtained, with $K \approx 0.5$, configuring the system with quasi-periodic behavior, as it is neither completely periodic nor chaotic. We can also observe that there is a region with chaotic behavior that also provides a good level of energy generation for values close to $\varepsilon_1 = 0.0904$ and $\rho = 0.0703$. For these parameters, we obtain an average power of energy $P_{avg} \approx 1.253$, with $K \approx 0.9$.

### 3.5. Dynamics and Potential of Energy Generation for Variation in Parameters $\varepsilon_2$ and $\theta$

Now, considering the case of varying the parameters $\varepsilon_2 = [1 : 10]$ versus $\theta = [0.01 : 0.4]$ and setting the parameters $\alpha_1 = 0.1$, $\Theta = 0.6$, and $\rho = 1$, we obtain the results shown in Figure 7.

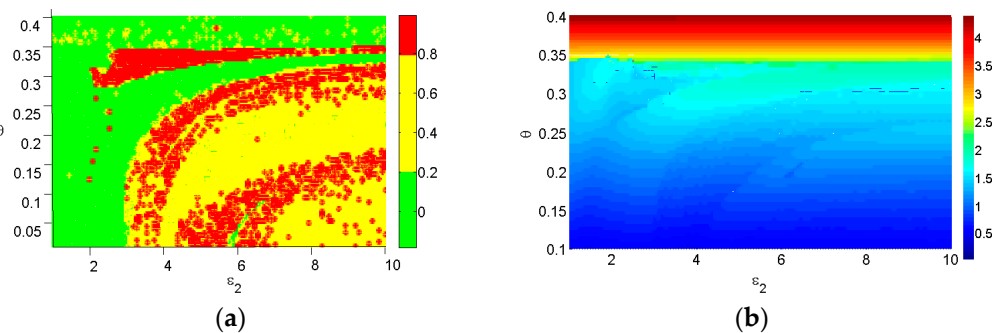

**Figure 7.** Variation in parameters $\varepsilon_2 = [1 : 10]$ versus $\theta = [0.01 : 0.4]$; (**a**) test 0–1; (**b**) average power.

The value of $\theta$ has a significant influence on energy generation, and $\varepsilon_2$ has an influence on the behavior of the system. Therefore, the greater the $\theta$, the greater the capacity of the system to generate energy, and the smaller the value of $\varepsilon_2$, the greater the possibility that the lateral movements of the structure have periodic behavior. In the case of using the parameters $\varepsilon_2 = 5$ and $\theta = 0.4$, an average power of energy $P_{avg} = 4.238$ is obtained with test value 0–1 $K \approx 0$.

### 3.6. Dynamics and Potential of Energy Generation for Variation in Parameters $\varepsilon_2$ and $\Theta$

Figure 8 shows the $K$ variations in the 0–1 test and the variations in the mean power for the case of $\varepsilon_2 = [1 : 10]$ versus $\Theta = [0.01 : 1.2]$, with $\varepsilon_1 = 1$, $\theta = 0.20$, and $\rho = 1$. An average power was obtained for these parameters.

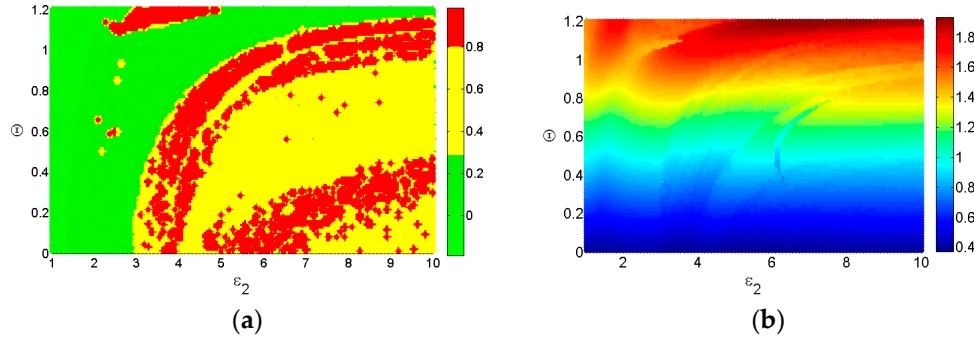

**Figure 8.** Variation in parameters $\varepsilon_2 = [1 : 10]$ versus $\Theta = [0.01 : 1.2]$: (**a**) test 0–1; (**b**) average power.

According to the results presented in Figure 8, the average power increases as the value of $\Theta$ increases, and the highest potential energy is found in the system's periodic behavior range. An average power energy of $P_{avg} = 1.930$ was obtained with $K \approx 0$ for $\varepsilon_2 = 5$ and $\Theta = 1.2$.

Another region with a significant energy potential value is obtained when $\varepsilon_2 = 3.364$ and $\Theta = 1.176$ are considered, providing an average potential energy $P_{avg} \approx 1.587$, with $K \approx 0.9$, indicating that the system will have chaotic behavior for this case.

### 3.7. Dynamics and Potential of Energy Generation for Variation in Parameters $\varepsilon_2$ and $\rho$

Figure 9 shows the dynamics of the system and the potential for power generation, considering variations in the parameters $\varepsilon_2 = [1 : 10]$ and $\rho = [0.01 : 2]$ and fixed parameters $\varepsilon_1 = 1$, $\theta = 0.20$, and $\Theta = 0.6$.

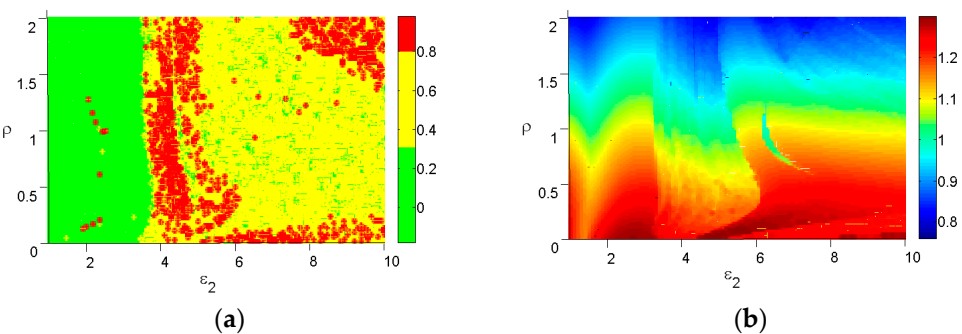

**Figure 9.** Variation in parameters $\varepsilon_2 = [1:10]$ versus $\rho = [0.01:2]$: (**a**) test 0–1; (**b**) average power.

Figure 9 shows that a higher energy generation value is obtained as the value of $\rho$ decreases, leading the system to quasi-periodic or periodic behavior. It was possible to obtain an average power energy of $P_{avg} = 1.299$ and $K \approx 0.4$ for the parameters $\varepsilon_2 = 4.909$ and $\rho = 0.0301$ and an average power of energy $P_{avg} = 1.291$ and $K \approx 0$ for the parameters $\varepsilon_2 = 6.636$ and $\rho = 0.01$.

### 3.8. Dynamics and Potential of Energy Generation for Variation in Parameters $\theta$ and $\Theta$

Now, consider the dynamics of the system and the potential for energy generation for variation in the parameters $\theta = [0.01:0.4]$ versus $\Theta = [0.01:1]$ with fixed parameters $\alpha_1 = 0.1$, $\alpha_2 = 0.1$, and $\rho = 1$.

Figure 10 presents the $K$ variations in the 0–1 test and the variations in the mean power.

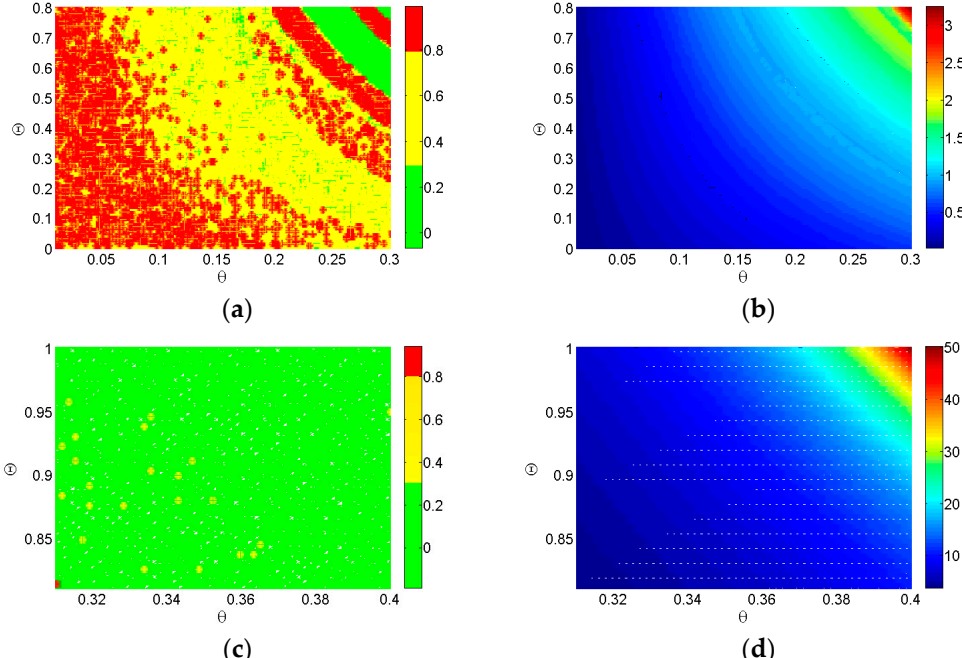

**Figure 10.** Variation in parameters: (**a**) test 0–1 for $\theta = [0.01:0.3]$ versus $\Theta = [0.01:0.8]$; (**b**) average power for $\theta = [0.01:0.3]$ versus $\Theta = [0.01:0.8]$; (**c**) test 0–1 for $\theta = [0.31:0.4]$ versus $\Theta = [0.81:1]$; (**d**) average power for $\theta = [0.31:0.4]$ versus $\Theta = [0.81:1]$.

As shown in Figure 10, energy generation increases as values of $\theta$ and $\Theta$ increase. The increase in the value of the parameters leads the system to periodic behavior. We can observe in Equation (3) that the parameters $\theta$ and $\Theta$ are related to the coupling of the piezoelectric material to the structure, so using the two parameters at maximum values may be physically impossible.

Consider the possibility of using the maximum values $\theta = 0.4$ and $\Theta = 1$, values that provide an average power expectation $P_{avg} \approx 49.844$ with $K \approx 0$.

### 3.9. Dynamics and Potential of Energy Generation for Variation in Parameters $\theta$ and $\rho$

The $K$ variations from the 0–1 test as well as the average power variations for $\theta = [0.01 : 0.4]$ versus $\rho = [0.01 : 2]$ for $\alpha_1 = 0.1$, $\alpha_2 = 0.1$, and $\Theta = 0.6$ are presented in Figure 11.

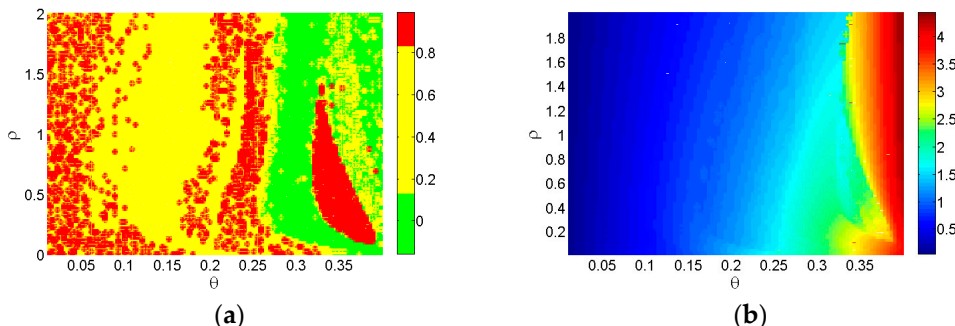

(a)  (b)

**Figure 11.** Variation in parameters $\theta = [0.01 : 0.4]$ versus $\rho = [0.01 : 2]$: (**a**) test 0–1; (**b**) average power.

Figure 11 shows that a higher energy generation value is obtained as the values of $\theta$ and $\rho$ increase, as well as the periodic behavior of the system, since we have an average power estimate $P_{avg} \approx 4.687$ with $K \approx 0$ for $\theta = 0.4$ and $\rho = 2$.

### 3.10. Dynamics and Potential of Energy Generation for Variation in Parameters $\Theta$ and $\rho$

In this subsection, the last combination of the parameters investigated in this paper is presented, considering variation in $\Theta = [0.01 : 1.2]$ and $\rho = [0.01 : 2]$, with parameters of the dynamics of the system and the potential for energy generation, considering the parameters $\alpha_1 = 0.1$, $\alpha_2 = 0.1$, and $\theta = 0.2$.

In Figure 12, we can observe the $K$ variation in the 0–1 test and the estimate of the average power generated by the system.

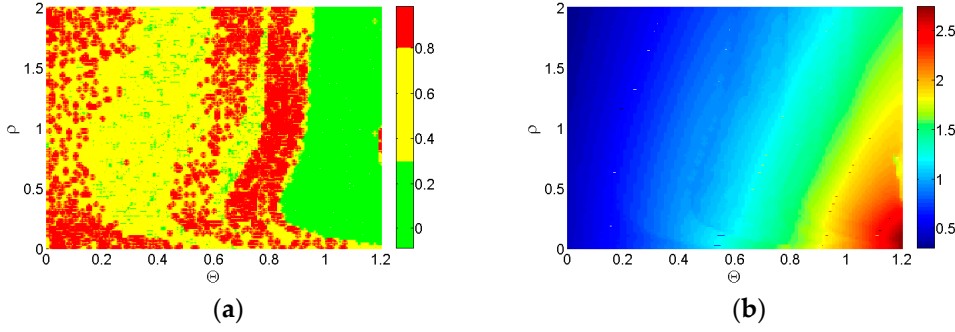

(a)  (b)

**Figure 12.** Variation in parameters $\Theta = [0.01 : 1.2]$ versus $\rho = [0.01 : 2]$: (**a**) test 0–1; (**b**) average power.

The results presented in Figure 12 show that a higher energy generation value is obtained as the value of $\Theta$ increases and $\rho$ decreases, and the system is kept in a region of periodic behavior. The highest estimated average power value is $P_{avg} = 2.687$ with $K \approx 0$ for the parameters $\Theta = 1.2$ and $\rho = 0.0904$.

## 4. Discussion

Table 1 presents a summary of the results of the average power variation, the value of the 0–1 test, and the classification of the system's behavior for variations in the control parameters, according to the results presented in the previous sections.

**Table 1.** Average power estimation and system behavior for parametric variation.

| Case | Parameter | Parameter | Average Power | K Value (Test 0–1) | Behavior |
|---|---|---|---|---|---|
| 1 | $\varepsilon_1 = 0.0502$ | $\varepsilon_2 = 1.182$ | $P_{avg} \approx 1.198$ | $K \approx 0.9$ | Chaotic |
| 2 | $\varepsilon_1 = 0.3919$ | $\varepsilon_2 = 1$ | $P_{avg} \approx 1.184$ | $K \approx 0$ | Periodic |
| 3 | $\varepsilon_1 = 0.0502$ | $\theta = 0.3961$ | $P_{avg} \approx 4.238$ | $K \approx 0$ | Periodic |
| 4 | $\varepsilon_1 = 0.01$ | $\Theta = 1.152$ | $P_{avg} \approx 1.587$ | $K \approx 0.9$ | Chaotic |
| 5 | $\varepsilon_1 = 0.2311$ | $\Theta = 1.188$ | $P_{avg} \approx 1.965$ | $K \approx 0$ | Periodic |
| 6 | $\varepsilon_1 = 0.03316$ | $\rho = 0.01$ | $P_{avg} \approx 1.355$ | $K \approx 0.5$ | Quasi-periodic |
| 7 | $\varepsilon_1 = 0.0904$ | $\rho = 0.0703$ | $P_{avg} \approx 1.253$ | $K \approx 0.9$ | Chaotic |
| 8 | $\varepsilon_2 = 5$ | $\theta = 0.4$ | $P_{avg} \approx 4.238$ | $K \approx 0$ | Periodic |
| 9 | $\varepsilon_2 = 3.364$ | $\Theta = 1.176$ | $P_{avg} \approx 1.587$ | $K \approx 0.9$ | Chaotic |
| 10 | $\varepsilon_2 = 5$ | $\Theta = 1.2$ | $P_{avg} \approx 1.93$ | $K \approx 0$ | Periodic |
| 11 | $\varepsilon_2 = 4.909$ | $\rho = 0.0301$ | $P_{avg} \approx 1.299$ | $K \approx 0.4$ | Quasi-periodic |
| 12 | $\varepsilon_2 = 6.636$ | $\rho = 0.01$ | $P_{avg} \approx 1.291$ | $K \approx 0$ | Periodic |
| 13 | $\theta = 0.4$ | $\Theta = 1$ | $P_{avg} \approx 49.844$ | $K \approx 0$ | Periodic |
| 14 | $\theta = 0.4$ | $\rho = 2$ | $P_{avg} \approx 4.687$ | $K \approx 0$ | Periodic |
| 15 | $\Theta = 1.2$ | $\rho = 0.0904$ | $P_{avg} \approx 2.687$ | $K \approx 0$ | Periodic |

In Table 1, we can observe the variation in the average power and behavior of the system for the following combinations of parameters: $\varepsilon_1 = [0.01 : 2]$, $\varepsilon_2 = [1 : 10]$, $\theta = [0.01 : 0.4]$, $\Theta = [0.01 : 1.2]$, and $\rho = [0.01 : 2]$.

We can observe that the predominance is for periodic behavior. The highest value observed for the system in periodic, quasi-periodic, and chaotic behavior is highlighted.

In Figure 13, we can observe the phase diagram of $x_1$ versus $x_2$, $x_3$ versus $x_4$, and the voltage variation in the piezoelectric material ($x_7$), considering the highest average power value for periodic behavior (case 13) obtained for the following parameters: $\alpha_1 = 0.1$, $\alpha_2 = 0.1$, $\alpha_3 = 0.5$, $\beta_1 = 1$, $\beta_3 = 0.2$, $\delta_1 = 8.373$, $\rho_1 = 0.05$, $\rho_2 = 100$, $\rho_3 = 200$, $\varepsilon_1 = 1$, $\varepsilon_2 = 5$, $\theta = 0.4$, $\Theta = 1$, and $\rho = 1$.

In Figure 14, we can observe the phase diagram of $x_1$ versus $x_2$, $x_3$ versus $x_4$, and the voltage variation in the piezoelectric material ($x_7$), considering the behavior as the highest average power estimate for the case of the system having chaotic behavior (case 4), considering the following parameters: $\alpha_1 = 0.1$, $\alpha_2 = 0.1$, $\alpha_3 = 0.5$, $\beta_1 = 1$, $\beta_3 = 0.2$, $\delta_1 = 8.373$, $\rho_1 = 0.05$, $\rho_2 = 100$, $\rho_3 = 200$, $\varepsilon_1 = 0.01$, $\varepsilon_2 = 5$, $\theta = 0.2$, $\Theta = 1.152$, and $\rho = 1$.

Analyzing the results presented in Figures 13 and 14, it is evident from the phase diagrams that there is periodic behavior for the first case and chaotic for the second case.

For the case $\theta = 0.4$ and $\Theta = 1$ (Figure 13c), the highest estimate of the average power was obtained. However, it is observed that the structure has a pronounced lateral displacement, and its oscillations are concentrated at a point far away from the origin (Figure 13c), which explains why the average power value is much higher than the others. As the voltage of the piezoelectric material depends on the deflection of the side frame of the portal frame, the flexed mantel will maintain a high voltage, and the small oscillations added to the deflection will only result in slight increases in the constant voltage. However, this behavior shifts the structure's center of gravity, generating the buckling effect.

Contrary to what is observed in Figure 13, in Figure 14 it can be observed that the side of the frame will oscillate around the origin, not displacing the side of its normal axis and thus maintaining the center of gravity at the base of the frame.

We can observe from the results presented in the previous section and Table 1 that the addition of the parameter $\theta$ has implied an increase in the average potency. However, it has also changed the structure's center of gravity, as seen in Figure 13.

In Figure 15, we present the phase diagram of $x_1$ versus $x_2$ for the other cases in which $P_{avg} > 4$, and it depends on the parameter $\theta$.

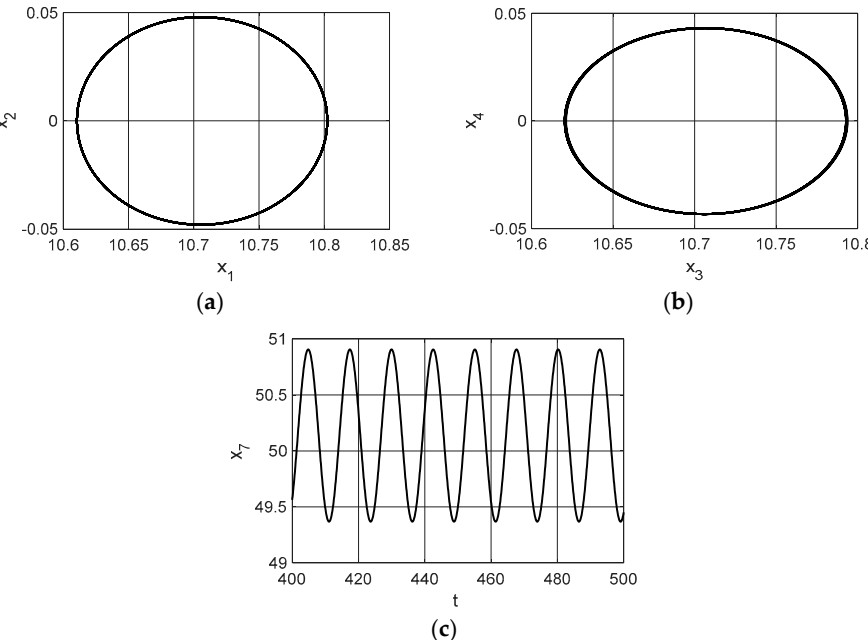

**Figure 13.** (**a**) Phase diagram $x_1$ versus $x_2$. (**b**) Phase diagram $x_3$ versus $x_4$. (**c**) Voltage variation in piezoelectric material $x_7$.

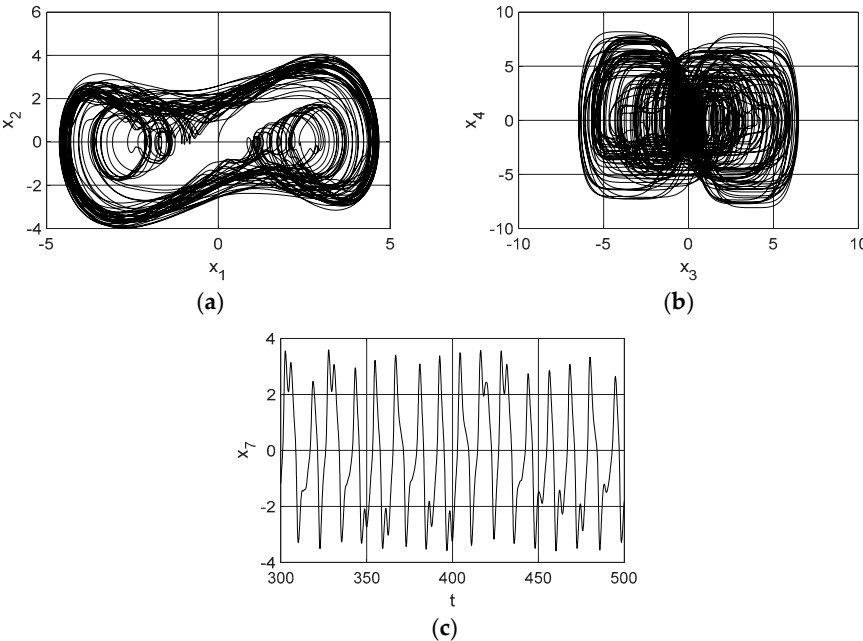

**Figure 14.** (**a**) Phase diagram $x_1$ versus $x_2$. (**b**) Phase diagram $x_3$ versus $x_4$. (**c**) Voltage variation in piezoelectric material $x_7$.

As can be seen in the phase diagrams of Figure 15, the increment of the parameter $\theta$, in addition to increasing the average power, also causes a permanent deflection in the portal frame column, changing the center of gravity of the gantry column in all cases.

As can be seen in Equation (3) the power $x_7$ depends on $((\theta + \theta\Theta|x_1|)x_1)$, as well as the displacement $x_1$ of $((\theta + \theta\Theta|x_1|)x_7)$, as we see the $\theta$ parameter is the coupling parameter between the structure and piezoelectric material that influences the energy more in relation

to the displacement and the rigidity of the structure; so, for $\theta > 1$ the power tends to increase even for small displacements of $x_1$, as well as the rigidity of the structure.

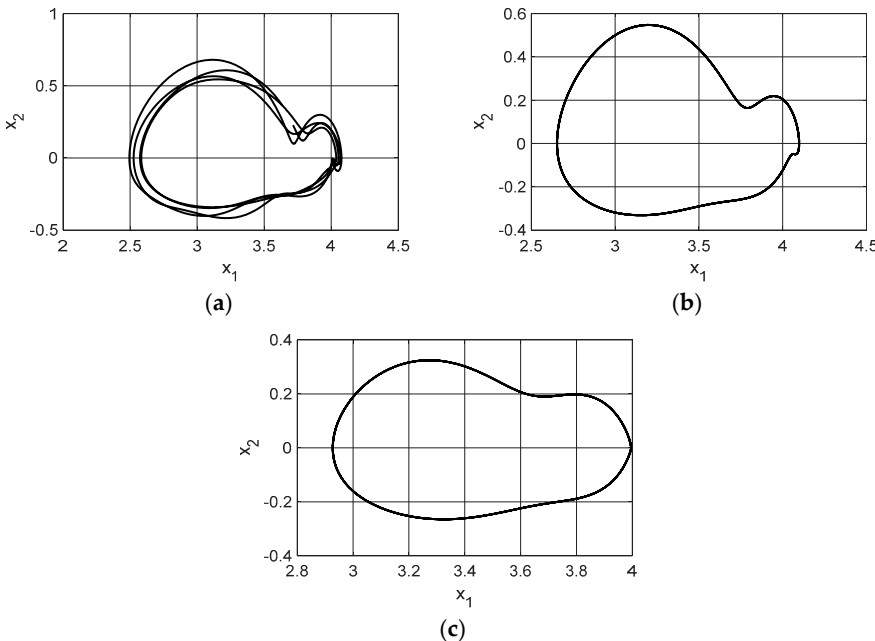

**Figure 15.** Phase diagram $x_1$ versus $x_2$: (**a**) case 3; (**b**) case 8; (**c**) case 14.

Considering the energy potential, we can also see in Table 1 that the parameter $\Theta$ is quite influential, as well as the parameter $\theta$.

Figure 16 shows the phase diagram of $x_1$ versus $x_2$ for the first three highest average powers, considering the dependence on the parameter $\Theta$ and not on $\theta$.

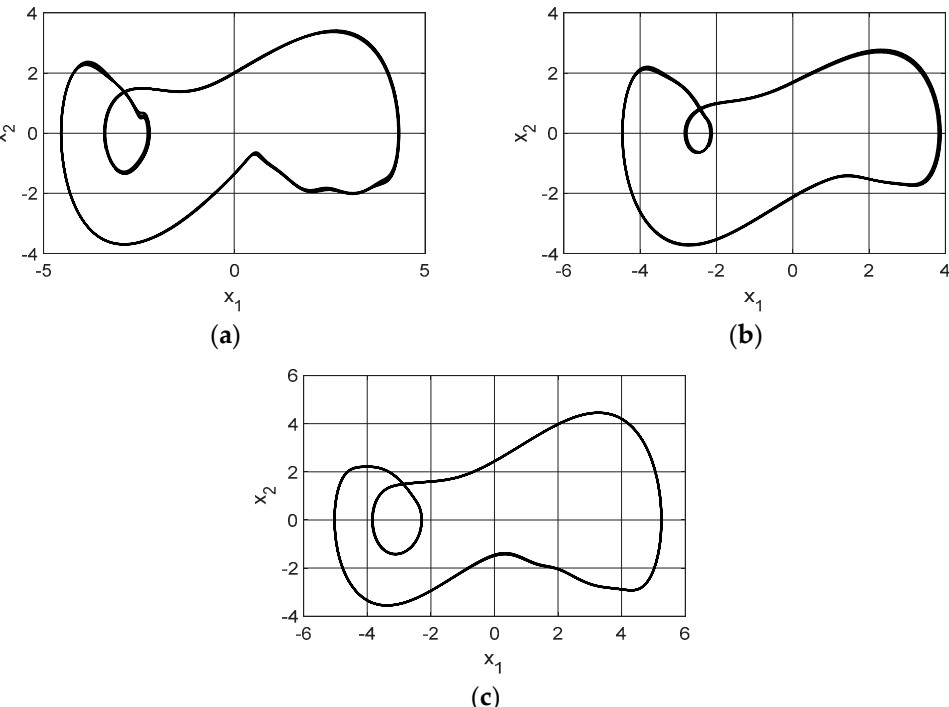

**Figure 16.** Phase diagram $x_1$ versus $x_2$: (**a**) case 5; (**b**) case 10; (**c**) case 15.

As seen in Figure 16, parameter $\Theta$ has little influence on the center of gravity of the portal frame column. Its variation will not influence cases of buckling. However, its effect on the behavior and estimation of the average power of the piezoelectric material is evident.

## 5. Conclusions

The presented results demonstrate the significant influence of the piezoelectric material parameters on the system's dynamic behavior. Since the piezoelectric material can be used both as an actuator and sensor, its use as a source of energy is essential for green energy harvesting. The stored energy, in turn, can be used later in applying the piezoelectric material as an actuator.

Considering the numerical results presented, we can highlight the influence of the linear coupling parameter "$\theta$" of the piezoelectric material. The results showed that the increase in the parameter's value could lead the structure to remain in deflection, which positively provides a more significant potential for generated energy; however, this action can leave the frame's column subject to undesirable effects, such as buckling. It was also seen that the increase in the value of the piezoelectric material's nonlinear coupling parameter "$\Theta$" also increases potential energy but without causing deflection.

With the 0–1 test, it was possible to verify which parameters can lead the system into chaotic behavior, which is not desired in most applications. The study also confirmed that the NES system is a good alternative for controlling vibrations without energy consumption.

Based on the contributions of numerical and dimensionless results presented, we can consider in future works the use of an optimization system through metaheuristics to determine the best combination of the five parameters simultaneously, as well as the assembly of an experimental apparatus for the analysis and validation of the model and the parameters used in dimensionless form, thus obtaining the average power in Watts.

**Author Contributions:** Conceptualization, A.M.T. and D.B.P.; methodology, M.E.K.F. and D.I.A.; software, A.M.T.; formal analysis, A.M.T. and G.G.L.; investigation, J.M.B.; writing—original draft preparation, A.M.T. and G.G.L.; writing—review and editing, M.E.K.F. and D.I.A.; visualization, J.M.B.; supervision, A.M.T. All authors have read and agreed to the published version of the manuscript.

**Funding:** This research received no external funding.

**Data Availability Statement:** Not applicable.

**Acknowledgments:** The authors thank the Capes, Fundação Araucária, and CNPq agency. The first author thanks CNPq for financial support (process: 310562/2021-0). The third author thanks CNPq for financial support (process: 309799/2021-0). The last author thanks CNPq for financial support (process: 304068/2022-5).

**Conflicts of Interest:** The authors declare no conflict of interest.

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
