# Peer review of "Dynamic Analysis and Piezoelectric Energy Harvesting from a Nonideal Portal Frame System including Nonlinear Energy Sink Effect"

_actuators, doi:10.3390/act12070298_

Round 1

Reviewer 1 Report

Summary: In the paper at hand, the authors analyze the dynamic performance and energy harvesting performance of a U-shaped portal frame-type system with an engine with unbalanced mass and coupled to a non-linear energy sink (NES). Numerical simulations are presented considering different combinations of parameters through 0-1 test and phase diagrams. This study has shown some significant potential for energy generation. However, I have a few queries/suggestions which the authors may please address.

(1)   The introduction focuses on the application prospects of piezoelectric materials, but does not present the background overview and advantages of nonlinear energy sink (NES) and the potential applications for energy harvesting in the frame-type system. These contents should be added.

(2)   In line 100, is “non-linear energy dissipator” the same as (NES) “nonlinear energy sink”? Please use the same one if they are same, including the abbreviation for easier reading.

(3)   Figure 1 is clearly missing some necessary descriptions. For example, the configuration of the NES, the specific meaning of NIS and the arrangement of piezoelectric layers all need to be described.

(4)   The physical parameters in figure 2 need to be explained.

(5)   A space is missing in each of subtitles 3.1, 3.2, 3.7, 3.9.

(6)   Extra space at the beginning of line 371.

(7)   The phenomenon in figure 13 is interesting, but why a small deformation produce such a large voltage? Please explain.

(8)   The parameter changed in cases 5,10,15 depicted in figure 15 is Θ, but why does line 361 say “it depends on the parameter θ”? And the same confusion appears in figure 16 which say “the dependence on the parameter Θ” but the parameter changed in cases 3,8,14 is θ. Please discuss.

Some problems in the equations:

(9)   In Eq (1), some parameters are not shown in the physical model (figure 2), such as kl, V1, V2, and these need to be explained.

(10) In line 117-120, the brackets in ω1 are in the wrong place and the format of fraction of ρ does not match the format of fractions of other variables, so it looks inconsistent.

(11) The derivative of the variables in the dimensionless equation with respect to the dimensionless time should be (·)', and the point is still used in Eqs. (2)-(3), but (·)' is used in Eq. (5).

(12) x(j) in Eqs (6)-(7) should be xj.

(13) Eqs (9)-(10) are written in a wrong way.

Some problems in the figures:

(14) All figures should be vectographs, not in the form of screenshots, and the coordinate name format should be uniform.

(15) The subscript figures in Figures (13)-(16) should be Roman type, not italic type.

see above

Author Response

We would like to thank the reviewers for spending their time in reading, reviewing, and commenting on our manuscript. Those comments are all valuable and very helpful for revising and improving our manuscript to a better scientific level.

We have studied the raised comments carefully and made corrections, which we hope that meet your requirements. Please, consider the reviewers' comments in black, the authors' answers in blue and the changes made to the paper in red. Attached.

Reviewer 2 Report

This manuscript investigates the application of piezoelectric materials in energy generation through numerical simulations. The influence of the piezoelectric material parameters used in the energy collection and the dimensioning parameters of the NES system was deeply analyzed in the paper. Numerical simulations are presented considering all combinations of the parameters of the piezoelectric material model and the NES considered. The specific comments are listed as below: (1) The structure of proposed energy harvester and the theoretical model are widely used. The innovation of this manuscript needs to be emphasized specifically. (2) What are the application scenarios of the energy harvester and its theoretical model proposed in this manuscript? (3) In section 2, the theoretical model of piezoelectric energy harvester was developed. It is suggested that this model be validated. (4) This manuscript conducted a large amount of dimensionless parameter influence analysis. However, the output voltage, current and output power of the energy harvester were not discussed, which is essential for energy harvesters. (5) The literature review is insufficient and need to be extended by including recently published works highly relate to the topic of this work. Here are some examples: Smart Materials and Structures, 2022, 31 (12): 125008; Applied Physics Letters, 2022, 121 (1): 013902. (6) There are some grammar and typo errors in the manuscript. Please check the manuscript and revise them. Overall, the work in this manuscript is beneficial for application of piezoelectric energy harvesting. This manuscript can be considered for publication after careful revision by incorporating the above comments. There are some grammar and typo errors in the manuscript. Please check the manuscript and revise them.

Author Response

We would like to thank the reviewers for spending their time in reading, reviewing, and commenting on our manuscript. Those comments are all valuable and very helpful for revising and improving our manuscript to a better scientific level.

We have studied the raised comments carefully and made corrections, which we hope that meet your requirements. Please, consider the reviewers' comments in black, the authors' answers in blue and the changes made to the paper in red. Attached

Reviewer 3 Report

This manuscript reported the numerical simulations, the application of piezoelectric  materials in energy generation. After carefully evaluating the manuscript, I recommend major revisions for the manuscript prior to publication in actuators.

Major comments:

1.      Author needs provide the experimental evidence to proof numerical simulations in the revised version of the manuscript.

2.      The real time application of the prepared piezoelectric nanogenerator is missing in the manuscript.

3.      The working mechanism of the prepared piezoelectric nanogenerator is missing in the manuscript.

4.      In the introduction part of manuscript is too general. Authors should include relevant literatures in the introduction section to emphasize the significance of their work. Some of the important literature needed to compare are listed below.

References:

[1]   J. Mater. Chem. A, 2019, 7, 21693-21703

[2]   Nano Research volume 14, pages 3669–3689 (2021)

[3]   Journal of Materials Science volume 57, pages4399–4440 (2022)

[4]   RSC Adv., 2017, 7, 33642-33670

[5]   Journal of Alloys and Compounds 918 (2022) 165653

This manuscript reported the numerical simulations, the application of piezoelectric  materials in energy generation. After carefully evaluating the manuscript, I recommend major revisions for the manuscript prior to publication in actuators.

Major comments:

1.      Author needs provide the experimental evidence to proof numerical simulations in the revised version of the manuscript.

2.      The real time application of the prepared piezoelectric nanogenerator is missing in the manuscript.

3.      The working mechanism of the prepared piezoelectric nanogenerator is missing in the manuscript.

4.      In the introduction part of manuscript is too general. Authors should include relevant literatures in the introduction section to emphasize the significance of their work. Some of the important literature needed to compare are listed below.

References:

[1]   J. Mater. Chem. A, 2019, 7, 21693-21703

[2]   Nano Research volume 14, pages 3669–3689 (2021)

[3]   Journal of Materials Science volume 57, pages4399–4440 (2022)

[4]   RSC Adv., 2017, 7, 33642-33670

[5]   Journal of Alloys and Compounds 918 (2022) 165653

Author Response

(The authors gave the same response as above.)

Round 2

Reviewer 2 Report

This work is well revised following reviewers' comments. It can be accepted for publication now.

Reviewer 3 Report

Accept it in its current form.